# Circular Economy Approach in Treatment of Galvanic Wastewater Employing Membrane Processes

**DOI:** 10.3390/membranes13030325

**Published:** 2023-03-11

**Authors:** Anna Kowalik-Klimczak, Anna Gajewska-Midziałek, Zofia Buczko, Monika Łożyńska, Maciej Życki, Wioletta Barszcz, Tinatin Ciciszwili, Adrian Dąbrowski, Sonia Kasierot, Jadwiga Charasińska, Tadeusz Gorewoda

**Affiliations:** 1Łukasiewicz Research Network—Institute for Sustainable Technology, 26-607 Radom, Poland; 2Łukasiewicz Research Network—Warsaw Institute of Technology, 00-661 Warsaw, Poland; 3Łukasiewicz Research Network—Institute of Non-Ferrous Metals, 44-121 Gliwice, Poland

**Keywords:** circular economy, ultrafiltration (UF), reverse osmosis (RO), galvanic wastewater, water recovery, zinc solution reuse

## Abstract

According to the idea of sustainable development, humanity should make every effort to care for the natural environment along with economic development. Decreasing water resources in the world makes it necessary to take action to reduce the consumption of this resource. This article presents the results of research conducted to improve the use of recyclable materials in line with the circular economy model. The research focused on the development of a technological solution for the recovery of raw materials from galvanic wastewater. The concept of a galvanic wastewater treatment system presented in the article includes wastewater pre-treatment in the ultrafiltration (UF) process and water recovery in the reverse osmosis (RO) process. In addition, the purpose of the work was to manage post-filtration waste (RO retentate) containing high concentrations of zinc in the process of galvanizing metal details. The obtained results indicate that it is possible to reduce the amount of sewage from the galvanizing industry by reusing the recovered water as technical water in the process line. The carried-out model tests of galvanizing confirmed the possibility of using RO retentate for the production of metal parts. The achieved results are a proposal to solve the problem of reducing the impact of galvanic wastewater on the environment and to improve the profitability of existing galvanizing technologies by reducing the consumption of water and raw materials.

## 1. Introduction

Wastewater generated by the galvanic industry primarily includes wastewater from rinsing and, occasionally, in a much smaller amount, from discharged solutions (baths) of main processes in a production electroplating line [1,2,3]. Discharging such types of wastewater into the environment is associated with meeting high administrative requirements. Therefore, appropriate treatment methods and techniques should be used, depending on the type of wastewater. Galvanic wastewater is usually divided into three streams. One of them is a stream that includes Cr(VI) compounds, which are chemically reduced to Cr(III) compounds. The second stream contains alkaline metal cyanides, which are oxidized and decomposed. After these treatments, these two streams are combined with a third acidic–alkaline stream. If an electroplating plant does not provide chromium plating based on Cr(VI) compounds or cyanide-based processes, there are only alkaline–acidic wastes to be treated. Alkaline–acidic wastes contain diluted heavy metal salts with usually acidic pH [4,5]. Nowadays, galvanic wastewater treatment most commonly employs a chemical method, which involves removing heavy metal ions by converting them into poorly soluble compounds (hydroxides or alkaline carbonates) using various alkaline reagents. This is a fairly simple and reliable method that offers easy automatic pH control. However, the amphoteric properties of metals make it difficult to properly define a pH range in which all heavy metal ions could precipitate at the same time. This method also involves high consumption of chemicals and the formation of reaction by-products in the form of residues, which constitutes its major disadvantage. Additionally, the presence of complexing agents in the solution makes metal extraction difficult [6,7].

In the case of galvanic wastewater characterized by low concentrations of metals and organic compounds, ion exchange proves effective. This method involves interaction between the wastewater and solid phase (an ion exchanger placed in specially designed columns) to remove heavy metal ions [8,9,10]. However, when such systems are used, the ion exchanger needs to be regularly regenerated (i.e., with acidic and base solutions), and the wastewater generated in this process must be disposed of for chemical neutralization.

Recently, electrocoagulation has also been used to treat galvanic wastewater. This process is similar to chemical coagulation, but in this method, the coagulant is produced directly in the reactor during the process reaction of ions released from metal electrodes under the influence of direct current between electrodes immersed in wastewater. This process destabilizes contaminant particles, using an electrical charge to hold them in the solution, which ensures the sorption of heavy metals on the surface of the resulting metal oxide and hydroxide. However, this method is of high electricity consumption, requires a constant concentration of contaminants in the wastewater stream, and impedes the precipitation of residues due to hydrogen release [10,11,12].

Alternatively, or to supplement the methods listed above, membrane filtration techniques can also be used to treat galvanic wastewater [13,14,15]. Such techniques, in accordance with the concept of the Best Available Techniques (BAT), are among the primary elements of clean (i.e., zero waste) technologies, ensuring that up to 60% of the treated water is recirculated and heavy metals are removed from wastewater. For this reason, these methods have been defined by the European Commission as a tool complying with the principles of the circular economy and enabling their effective implementation [16,17]. The use of membrane processes, including micellar-enhanced ultrafiltration (MEUF) [18], polymer-enhanced ultrafiltration (PEUF) [19,20], reverse osmosis (RO) [21], and nanofiltration (NF) [22], offers interesting possibilities in regard to the removal of heavy metal ions from galvanic wastewater. However, RO and NF processes need to be preceded by wastewater pre-treatment that can employ classic chemical precipitation methods, which unfortunately involve the generation of sludge and residues that have to be disposed of. On the other hand, the effectiveness of metal ion release from galvanic wastewater in the NF process heavily depends on the concentration of salts of univalent and multivalent ions [22,23,24]. Therefore, a combined UF/RO system is far more effective when it comes to galvanic wastewater treatment [25]. Such a combination of membrane processes enables recovery of the treated water, which can be reused in production cycles of a galvanizing plant and reduce the amount of industrial wastewater contaminated with heavy metals [21].

The available literature in the field of galvanic wastewater treatment is focused on the use of membrane techniques primarily for water recovery. However, no data is available on the use of residue after filtration. The undertaken research works meet the need to use hazardous post-filtration streams as secondary raw materials in technological processes in galvanizing plants. The subject of the research was wastewater from the process of rinsing galvanized metal details in rotating drums in a low-acidic chlorine bath containing organic shining agents. A concept of galvanic wastewater treatment technology was proposed, enabling water recovery with the simultaneous use of the generated by-products. As part of the work, model tests of zinc coating deposition on metal elements were carried out, along with the selection of process parameters. The properties of these elements were compared with the properties of galvanized elements in a reference bath.

## 2. Materials and Methods

### 2.1. Galvanic Wastewater Characteristics

The authors performed a multiparameter physical and chemical analysis to define the characteristics of the galvanic wastewater in the form of (i) wastewater from a low-acidic chlorine bath containing organic shining agents and (ii) streams of post-membrane filtration liquids. The conductivity and pH of the liquid were measured with a Mettler Toledo Seven Excellence benchtop meter (Shah Alam, Malaysia). The turbidity was measured with a HACH 2100Q IS portable meter (Loveland, CO, USA). The chemical oxygen demand was determined via a LAR QuickCoDlab analyzer (Marseille, France). Measurements were based on high-temperature sample combustion at 1200 °C to ensure high accuracy and reliability of readings. The total carbon and total nitrogen-bound concentrations were determined via an Elementar vario TOC cube analyzer (Langenselbold, Germany). The analysis involved the measurement of carbon dioxide emitted as a result of high-temperature catalytic combustion of the sample exposed to an oxygen stream. The concentration of chlorides and sulfates was determined via the LCK cuvette tests and a Hach Lange UV-VIS DR 6000 spectrophotometer (Düsseldorf, Germany). The dry matter was determined via a moisture analyzer based on the difference in mass before and after drying at 105 °C. The boron and potassium concentrations were determined via a Perkin Elmer Optima 5300 DV (Markham, ON, Canada) inductively coupled plasma optical emission spectrometer (ICP-OES) (Tokyo, Japan). The zinc concentration was determined via an Analityk Jena NovAA 400 flame atomic absorption spectrometer (FAAS) (Munich, Germany). Before chemical elements were determined, the liquid samples were mineralized in nitric acid(V) using an Anton Paar Multiwave PRO pressure mineralizer (Warszawa, Poland).

### 2.2. Membrane Filtration

Membrane processes were conducted via an integrated membrane system to recover water and/or raw material from industrial wastewater (Figure 1). The system used is composed of four pilot process lines which are equipped with control systems allowing free process integration and are intended for the following four processes employing spiral wound polymer membranes: (1) microfiltration (MF); (2) ultrafiltration (UF); (3) nanofiltration (NF); and (4) reverse osmosis (RO).

To treat wastewater from a low-acidic chlorine bath containing organic shining agents, an integrated UF/RO system (Figure 2) was used. First, the wastewater was placed in tank 1 and then pumped to the UF module, where it was separated into permeate, fed to tank 2, and retentate redirected back to tank 1. The second process phase involved transferring the post-UF permeate to the RO module, where it was then separated into permeate, fed to tank 3, and retentate used in model galvanization tests. The initial volume of the galvanic wastewater was 100 dm^3^. The UF was carried out until a five-fold reduction in the feed was achieved (i.e., until 80 dm^3^ of the permeate was removed). On the other hand, the RO was carried out on the post-UF permeate until a four-fold reduction in the feed was achieved (i.e., until 60 dm^3^ of the permeate was removed).

The basic parameters for the UF and RO processes obtained via the commercial membrane modules are presented in Table 1.

The performance of membrane processes was assessed on the basis of the permeate flux (*J_P_*, dm^3^/(m^2^h)):(1)$$J_{P}=\frac{V_{P}}{A\times t}$$
where *V_P_*—permeate volume (dm^3^), *A*—membrane area (m^2^), and *t*—time needed to obtain specific permeate volume (h).

On the other hand, the effectiveness of the contaminant removal during membrane processes was assessed on the basis of the reduced content (%) of contaminants in the solution, i.e., based on the retention factor (*R*, % m/m):(2)$$R=\left(1-\frac{C_{1}}{C_{2}}\right)\times 100\%$$
where *C*_1_—component concentration in the liquid after treatment (mg/dm^3^), and *C*_2_—component concentration in the liquid before treatment (mg/dm^3^).

### 2.3. Mode-Galvanizing Tests

To assess whether the retentate obtained after the RO process from the wastewater from a low-acidic chlorine bath containing organic shining agents can be reused, the galvanizing process had to be carried out in accordance with all the substrate preparatory treatment steps. First, the retentate was subjected to quantitative analysis (titration), and then the levels of chloride bath components were replenished to the required concentrations to ensure proper bath performance. Coatings were produced on a laboratory scale by means of electrodeposition from the zinc bath solution prepared in this way. Tests were conducted in a Hull cell to select proper coating deposition parameters. This method allows the determination of the optimum range of the current density values, for which a good quality coating with a wide gloss range is obtained. Laboratory tests were performed in 0.5 cm^3^ of the bath based on RO retentate. Zinc coatings were deposited on a 4.0 × 2.0 cm carbon steel substrate and on industrial details (steel nails with a length of 10.0 cm and a diameter of 0.45 cm). The substrates were pre-treated in an electrolytic bath for degreasing the cathodic process. The process lasted until a continuous film of water was observed on the surface after rinsing. Then, details were activated in a 1:1 hydrochloric acid solution. This was followed by rinsing the sample and applying the zinc coating at a constant current density of 1.2 A/dm^2^ using magnetic stirring (100 rpm) for 1 h. After this process, the details were rinsed again, passivated in a chromium bath containing chromium(III) compounds, rinsed once more, and then dried. Distilled water was used at all rinsing stages. For comparison, analogous coating samples were also prepared in a tap water-based reference bath. Coating thickness measurements on samples from the retentate-based and tap water-based reference baths were carried out using an X-ray fluorescence spectrometer (FISCHERSCOPE X-RAY XDV-SDD, Tokye, Japan). Surface morphology and composition were analyzed using a ZEISS scanning electron microscope (Joel) (Oberkochen, Germany) and a KEYENCE VHX 5000 digital optical microscope (Itasca, IL, USA). SEM images were captured at 3000 V and WD of 4.6–4.7 mm. Surface roughness measurements were taken with a SURFTEST SJ-210 profilograph. The arithmetic average height parameter (Ra) is defined as the average absolute deviation of the roughness irregularities from the mean line over one sampling length [26]:(3)$$Ra=\frac{1}{l}\int_{0}^{l}\left|y\left(x\right)\right|dx$$
where *l*—relative length of the profile (-).

Ten-point height parameter (Rz) is defined as the difference in height between the average of the five highest peaks (*p_i_*) and the five lowest valleys (*v_i_*) along the assessment length of the profile [26]:(4)$$Rz=\frac{1}{n}\left(\sum_{i=1}^{n}p_{i}-\sum_{i=1}^{n}v_{i}\right)$$
where *n*—number of samples along the assessment length (-).

Microhardness by the Vickers method was measured with a WILSON-HARDNESS hardness tester (BUEHLER) (Warszawa, Poland) for the load of 0.025 kg.

## 3. Results and Discussion

### 3.1. Galvanic Wastewater Treatment in the UF/RO System

The results of the multiparameter analysis of wastewater from a low-acidic chlorine bath containing organic shining agents show that this type of galvanic wastewater is characterized by acidic pH, high salinity, and high heavy metal content (Table 2). The results presented in Table 2 indicated that wastewater should be pre-treated in the UF process, which will enable the separation of colloidal substance particles, and, according to Qin et al. [27], the appropriate UF pre-treatment might reduce RO membrane fouling and increase the efficiency of the removal process. Wastewater prepared in this way should be subjected to RO, which will enable its desalination and concurrent concentration of metal ions.

The UF was carried out under a pressure of 4.0 bar, and for the feed flow of 400 dm^3^/h, it was characterized by an average performance of 53.8 ± 8.9 dm^3^/(m^2^h) for a UP150 membrane. The applied UF membrane enabled contaminants responsible for turbidity and total suspended solids to be removed from the galvanic wastewater, and it helped reduce the concentration of organic compounds (Figure 3a). After UF, the permeate was subjected to RO. The RO process was carried out under the pressure of 30 bar, and for the feed flow of 150 dm^3^/h, it was characterized by the average performance of 6.9 ± 1.1 dm^3^/(m^2^h) for a TM710 membrane. The authors found that the UF/RO system enabled the removal of the vast majority of all analyzed galvanic wastewater components (Figure 3b). It should be noted that the RO membrane used allowed significant retention of chlorides (99%), zinc (99%), potassium (99%), and boron (86%). Additionally, no significant decrease in either membrane performance was observed during the process. During UF, a decrease in performance of 30% was observed for a five-fold reduction in the initial wastewater volume. During RO (carried out on wastewater pre-treated in the ultrafiltration process), a decrease in performance of 25% was observed for a four-fold reduction in the wastewater volume.

The results summarized in Table 3 indicated that the use of an integrated UF/RO system is a rational solution to treat galvanic wastewater in the form of a used rinse bath. The application of such a membrane system enables liquids to be effectively recovered and reused by a galvanizing plant. Similar results were presented by Petrinic et al. [25]. The authors show that the UF/RO system removed the contaminants from the galvanic wastewater, such as metal elements and organic and inorganic compounds. Contaminants were not completely removed from galvanic wastewater, but concentrations in the permeate were at low levels; thus, the quality of the permeate met the reuse criteria. Based on an economic analysis [25], it was concluded that the proposed UF/RO system is applicable within the metal finishing industry and also has the potential for use in others industries with similar water volumes and types of wastewater contaminants.

However, when membrane processes are used to treat galvanic wastewater, retentate streams in the form of concentrated contaminants retained by the membrane are generated. Despite a number of studies on membrane treatment of galvanic wastewater [4,21,25], research about retentate utilization is lacking. In the proposed technological solution for galvanic wastewater treatment, the RO process generates retentate containing zinc, potassium, boron, and chloride ions (Table 3). This composition of the RO retentate makes it possible perspective to reuse it in the industry.

### 3.2. Tests Verifying the Possibility to Use Retentate after RO

In the study, two types of media were used to prepare process bath solutions, i.e., (i) tap water and (ii) RO retentate. The retentate was analyzed for the basic components of the galvanizing solution. The concentrations of zinc, potassium chloride, and boric acid were determined. The bath was replenished to ensure that the concentration levels were comparable with an industrial chloride bath for galvanizing. The influence of the addition of shining agents on the appearance of the samples in terms of color and gloss was examined. Based on the Hull cell test results, zinc coating deposition parameters were determined (Table 4). Three variants of solutions were tested, (i) without additives, (ii) with a gloss carrier, and (iii) with two commercial shining agents, with appropriate concentrations selected in accordance with the reagent manufacturer’s formula.

Based on the test results, a bath containing a shining agent and a gloss carrier was selected. The optimum deposition process parameters were also determined, including a current density value of 1.2 A/dm^2^. The result of the Hull cell test proved that this bath also met the requirements for the appearance of the sample in terms of color and gloss. The appearance of the samples obtained after dipping in RO retentate and a tap water-based bath is shown below. The samples were produced on a stainless steel substrate (Figure 4a) and industrial details (Figure 4b).

The images confirm the validity of the parameters established in the Hull cell study. The obtained samples are shiny and have a bright metallic color. The samples were subjected to thickness measurement using an X-ray fluorescence spectrometer, which results are presented in Table 5.

The values of the zinc coating thickness measurements for the samples deposited from the retentate- and tap water-based baths are comparable. This means that both baths allow for the production of comparable coatings under the same conditions. The thickness test makes it possible to conclude that both the retentate- and tap water-based baths used allow the deposition of coatings with similar thicknesses. The surface morphology of the zinc coatings was analyzed using digital optical microscopy (Figure 5) and SEM (Figure 6).

Based on the optical microscope and SEM images (Figure 5 and Figure 6), the coatings produced in both types of baths were found to have a low degree of surface development. However, the zinc coating obtained from the RO retentate-based bath reflects the subsurface condition, as presented in Figure 5I. In contrast, for the sample obtained from the tap-water bath, this effect is not visible. This might be due to the quality of the substrate preparation treatment carried out prior to the process. The SEM images indicate a similar surface morphology, although the occurrence of coating inclusions/defects on samples from the RO retentate-based bath should be noted. Nevertheless, these differences are not significant, indicating that both zinc baths used in this study allow good-quality coatings to be obtained. In addition, the results of the roughness tests for the zinc coatings are presented in Table 6. The roughness values for the coating deposited in the RO retentate-based bath are higher compared to the roughness values of the coating produced in the reference bath. This can be due to the presence of impurities in the RO retentate, however, having a minor effect on the rest of the coating parameters. In fact, the low roughness values for both coating types confirm the low degree of surface development.

To assess the quality of the coatings, HV microhardness tests were performed (Table 7). The microhardness values obtained, including the standard deviation, were equal for both coatings types. This shows that the bath medium used had no impact on the microhardness of the coating obtained.

The use of the RO retentate to prepare galvanizing bath makes it possible to obtain good quality zinc coatings (in terms of appearance, thickness, surface morphology, microhardness, and roughness). The obtained research results on the properties of coatings are consistent with the literature data and previous investigations carried out in our laboratories [28,29]. Therefore, the resulting retentate can be used in zinc plating processes under certain production conditions. However, additional studies on a production scale must be carried out to verify whether the retentate can be used in an industrial galvanizing line.

## 4. Conclusions

The developed concept of galvanic wastewater treatment technology assumes the use of an integrated system of UF and RO processes for the recovery of liquids for reuse. The use of UF as a pre-treatment step allowed the removal of more than 90% of solid impurities. The RO process, on the other hand, allowed the removal of chlorides, zinc, potassium, and boron with an efficiency of 86–99%. By application of the UF/RO system, the final permeate stream can be reused as technical water. In turn, the final retentate stream was used for galvanizing metal elements. As part of the research, the parameters of the deposition of zinc coatings and the composition of the galvanizing bath were developed. On the basis of the tests carried out in the Hull electrolytic cell, it was determined that the optimal current density is 1.2 A/dm^3^ at a current intensity of 1.0 A. However, the tests of the elemental composition of the retentate showed that the bath based on it should be supplemented with commercial components to the required concentration values and additions of shining and gloss agents. The properties of coatings deposited from a bath based on retentate were compared with the properties of coatings produced from a reference bath based on tap water. The obtained results indicate that the properties of both compared zinc coatings are similar. Surface morphology studies showed the presence of small defects in the coating from the retentate-based bath. However, these differences are not significant, which is confirmed by thickness, roughness, and microhardness tests. This allows concluding that the obtained RO retentate can be reused for the deposition of good-quality zinc coatings. In the next stage of work, tests are planned for verification of the possibility of using the developed treatment system in real operating conditions directly in an industrial plant.

## Figures and Tables

**Figure 1 membranes-13-00325-f001:**
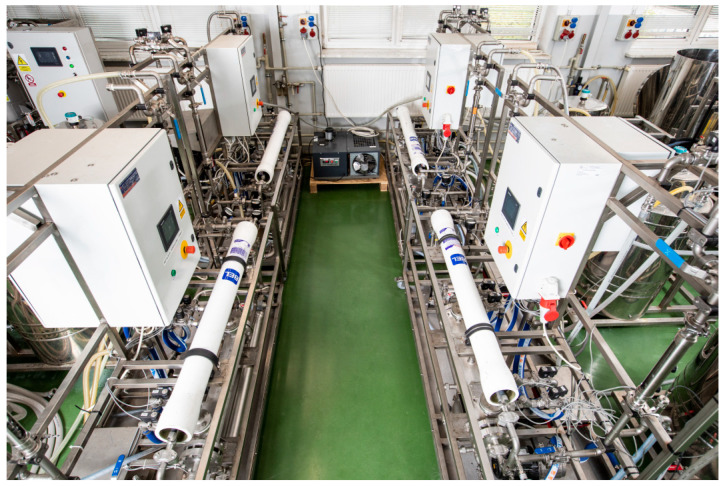
Integrated membrane system for industrial wastewater treatment.

**Figure 2 membranes-13-00325-f002:**
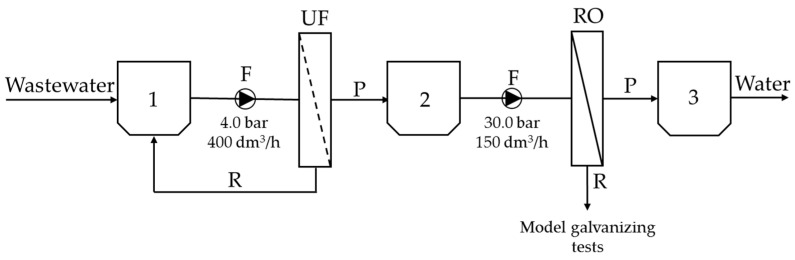
Technological diagram of the integrated membrane system: UF—ultrafiltration; RO—reverse osmosis; 1, 2, 3—process tanks; F—feed; P—permeate; R—retentate.

**Figure 3 membranes-13-00325-f003:**
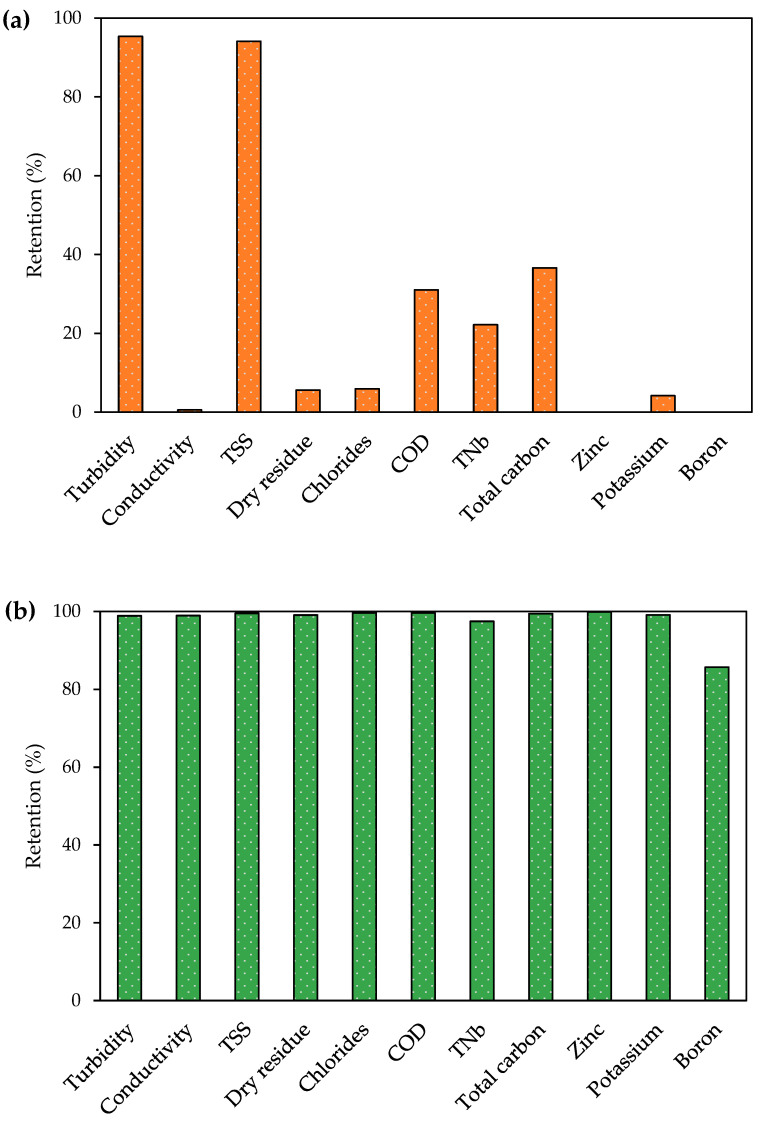
Retention of individual galvanic wastewater components in (**a**) UF and (**b**) UF/RO processes.

**Figure 4 membranes-13-00325-f004:**
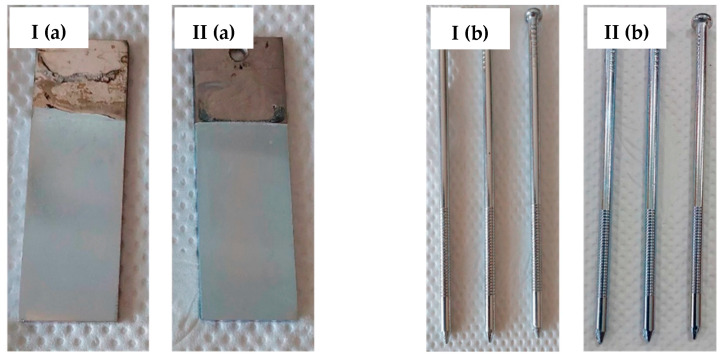
Appearance of zinc coating deposited on a flat carbon steel substrate (**a**) and industrial details (**b**) from baths prepared on the basis of the RO retentate (**I**) and tap water (**II**).

**Figure 5 membranes-13-00325-f005:**
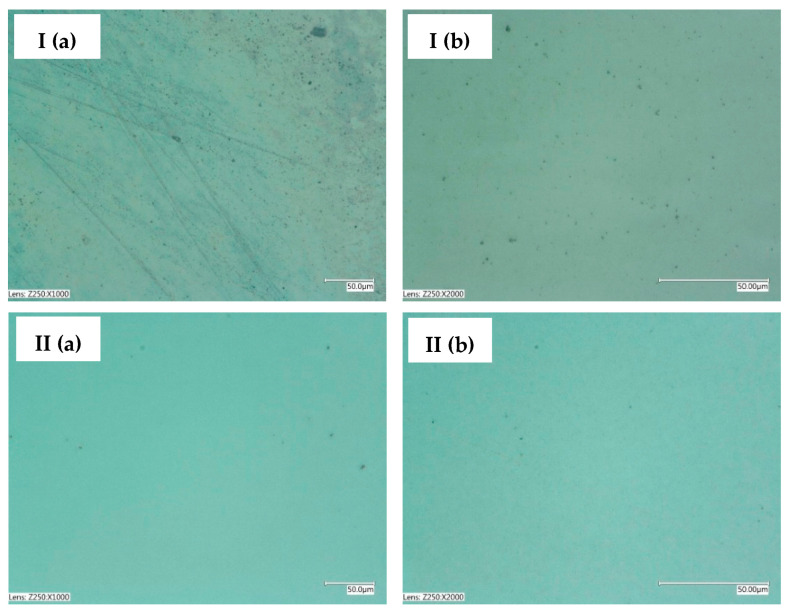
Surface morphology presented in optical microscope images ((**a**)—magnification ×1000, (**b**)—magnification ×2000) of zinc coatings obtained from (**I**) the RO retentate-based bath and (**II**) tap water-based bath.

**Figure 6 membranes-13-00325-f006:**
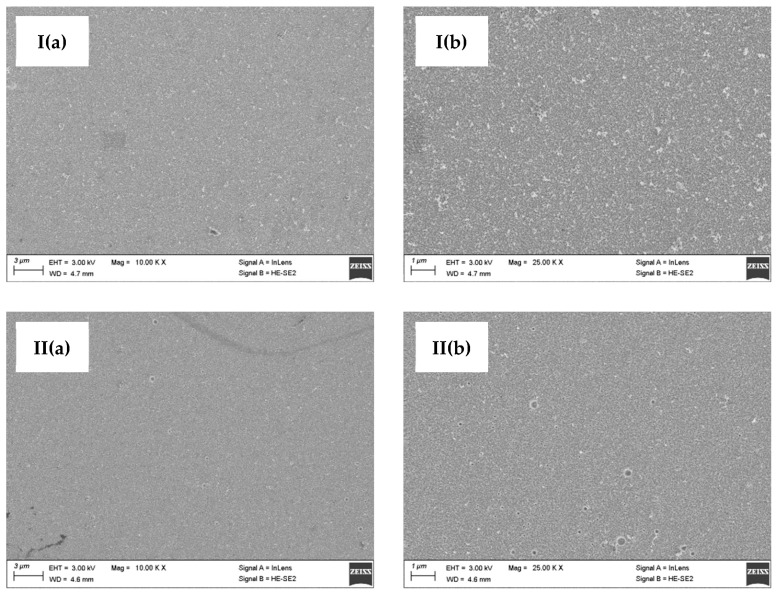
Surface morphology showed in SEM images ((**a**)—magnification ×10,000, (**b**)—magnification ×25,000) of zinc coatings obtained from (**I**) the RO retentate-based bath and (**II**) tap water-based bath.

**Table 1 membranes-13-00325-t001:** Parameters of membranes used to treat galvanic wastewater.

Type of Membrane	UF UP150	RO TM710
Manufacturer	Microdyn Nadir	Toray
Material	PES	PA
Area (m^2^)	6.0	8.1
NaCl retention (%)	-	99.7
Cut-off (Da)	150,000	-
pH range	0–14	2–11
Max. temperature (°C)	95	45

PES—polyethersulfone; PA—polyamide.

**Table 2 membranes-13-00325-t002:** Physical and chemical parameters of the galvanic wastewater.

Parameter	Value
pH	4.121 ± 0.004
Turbidity (FNU)	420.3 ± 1.5
Conductivity (mS/cm)	35.919 ± 0.036
Total suspended solids (mg/dm^3^)	431.3 ± 7.2
Dry residue (%)	3.289 ± 0.011
Chlorides (g/dm^3^)	11.042 ± 0.018
Chemical oxygen demand (g O_2_/dm^3^)	15.95 ± 0.21
Total nitrogen bound (mg/dm^3^)	65.8 ± 3.7
Total carbon (g/dm^3^)	4.256 ± 0.072
Zinc (g/dm^3^)	2.802 ± 0.013
Potassium (g/dm^3^)	9.560 ± 0.010
Boron (mg/dm^3^)	235.1 ± 1.6

**Table 3 membranes-13-00325-t003:** Physical and chemical parameters of the permeate and retentate after RO.

Parameter	Permeate	Retentate
pH	5.145 ± 0.003	4.146 ± 0.005
Turbidity (FNU)	2.507 ± 0.093	217.7 ± 2.1
Conductivity (mS/cm)	0.6421 ± 0.0018	58.71 ± 0.20
Total suspended solids (mg/dm^3^)	0.983 ± 0.029	192.7 ± 1.5
Dry residue (%)	0.052 ± 0.002	5.541 ± 0.039
Chlorides (g/dm^3^)	0.0759 ± 0.0015	19.750 ± 0.370
Chemical oxygen demand (mg O_2_/dm^3^)	<LOD	24,927 ± 76
Total nitrogen bound (mg/dm^3^)	2.552 ± 0.069	99.9 ± 2.1
Total carbon (g/dm^3^)	0.03694 ± 0.00047	6.050 ± 270
Zinc (g/dm^3^)	0.00650 ± 0.00010	4.800 ± 0.010
Potassium (g/dm^3^)	0.160 ± 0.005	17.10 ± 0.17
Boron (mg/dm^3^)	51.20 ± 0.72	357.0 ± 4.3

**Table 4 membranes-13-00325-t004:** Optimization of process conditions and bath composition for zinc coating using Hull cell test.

Conditions of Deposition	Bath Composition	Appearance of the Deposited Coating
I = 1.0 A current density 0.1–5.0 A/dm^2^	ZnCl_2_ KCl H_3_BO_3_	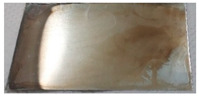 dark and matt
	ZnCl_2_ KCl H_3_BO_3_ gloss carrier	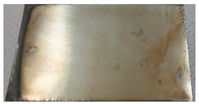 dark and matt
	ZnCl_2_ KCl H_3_BO_3_ gloss carrier gloss additive	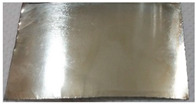 bright and glossy

**Table 5 membranes-13-00325-t005:** Results of thickness measurements of zinc coatings.

Sample	Thickness (μm)	
	Samples from the RO Retentate-Based Bath	Samples from the Tap Water-Based Bath
Flat substrate	16.9 ± 0.8	17.2 ± 0.6
Industrial detail No. 1	16.5 ± 0.8	14.5 ± 1.8
Industrial detail No. 2	17.1 ± 0.6	17.3 ± 1.8
Industrial detail No. 3	16.9 ± 1.0	16.5 ± 0.9

**Table 6 membranes-13-00325-t006:** Results of measurements of roughness parameters Ra and Rz of zinc coatings obtained from two types of baths.

Type of Coating	Roughness (μm)	
	Ra	Rz
Coating deposited in the RO retentate-based bath	0.023 ± 0.007	0.186 ± 0.042
Coating deposited in the tap water-based bath	0.016 ± 0.001	0.124 ± 0.018

**Table 7 membranes-13-00325-t007:** Results of measurements of microhardness (HV 0.025) of zinc coatings obtained from two types of baths.

Type of Coating	Microhardness (HV 0.025)
Coating deposited in the RO retentate-based bath	127 ± 3
Coating deposited in the tap water-based bath	125 ± 3

## Data Availability

Not applicable.

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
