# Peer review of "Circular Economy Approach in Treatment of Galvanic Wastewater Employing Membrane Processes"

_membranes, 2023, doi:10.3390/membranes13030325_

Round 1

Reviewer 1 Report

This study aimed to manage galvanic wastewater using ultrafiltration (UF) process and reverse osmosis (RO) process. With a pilot-scale experiment test, extra water is recovered from the combined process and the RO retentate was successfully utilized. This approach will help reduce the amount of wastewater from the galvanizing industry and produce additional water for the industry. While the concept is interesting, its academic benefits are not extensively included. Address the comments to improve the quality of the paper.

Comments:

1.        Introduction: The last paragraph should clearly state the scope of the study. It is difficult to understand the objective and scope of the study. Revise this.

2.        Fig. 3: While UF and RO are only utilized for the study, MF and NF are included in the figure. The process diagram should be revised according to the content of the study.

3.        Unit: dm3 is not frequently used. It will be better to use “L” or “m3” instead.

4.        Line 219: UF might not separate dissolved particles. Revise the sentence after consideration.

5.        Line 233: Operating flux for RO is too low as 7 LMH. Note that SWRO and BWRO are operated as 12-14 and 20-40 LMH, respectively.

6. Section 3.2: The section is not included.

7.        Table 3: Conductivity of permeate is higher than retentate. Check the information.

8.        Process diagram: It is recommended to include a whole process diagram including information of flow rate and concentration.

Author Response

20.02.2023

Dr inż. Anna Kowalik-Klimczak

Response on review 1

about manuscript:

„Circular Economy Approach in Treatment of Galvanic Wastewater Employing Membrane Processes”, Anna Kowalik-Klimczak, Anna Gajewska-Midziałek, Zofia Buczko, Monika Łożyńska, Maciej Życki, Wioletta Barszcz, Tinatin Ciciszwili, Adrian Dąbrowski, Sonia Kasierot, Jadwiga Charasińska, Tadeusz Gorewoda

Thank you very much for the valuable suggestion and insightful comments contained in the review. Answers to the comments are presented below:

Ad. 1. Introduction: The last paragraph should clearly state the scope of the study. It is difficult to understand the objective and scope of the study. Revise this.

Author response:

As suggested by the reviewer, the last paragraph in the Introduction section has been changed.

Ad. 2. Fig. 3: While UF and RO are only utilized for the study, MF and NF are included in the figure. The process diagram should be revised according to the content of the study?

Author response:

Fig. 3 (now fig. 2) has been corrected in accordance with the comments.

Ad. 3. Unit: dm3 is not frequently used. It will be better to use “L” or “m3” instead

Author response:

Thank you for your comment, however, the authors tried to use SI units (cm3, dm3).

Ad. 4. Line 219: UF might not separate dissolved particles. Revise the sentence after consideration

Author response:

Thanks for the suggestion, of course it was a mistake. The sentence has been corrected.

Ad. 5. Line 233: Operating flux for RO is too low as 7 LMH. Note that SWRO and BWRO are operated as 12-14 and 20-40 LMH, respectively.

Author response:

For the tested RO membrane, the permeate flux for demineralized water was 14 LMH at a pressure of 30 bar. During the RO process carried out on pre-treated galvanic wastewater, it was observed that the permeate flux is 50% lower. This is due to the high concentration of components, primarily chlorides and metal ions, in the feed liquid for the RO process.

Ad. 6. Section 3.2: The section is not included.

Author response:

Chapter numbering has been corrected.

Ad. 7. Table 3: Conductivity of permeate is higher than retentate. Check the information

Author response:

Thank you for your suggestion, all results have been checked and corrected.

Ad.8. Process diagram: It is recommended to include a whole process diagram including information of flow rate and concentration.

Author response:

As suggested by the reviewer, the process diagram was corrected.

Reviewer 2 Report

This manuscript proposed a integrate membrane process for the treatment of galvanic  wastewater. It was a attractive work for the industrial application of membrane process. However, much more details and investigation should be presented in the manuscript. For example, the fouling the flux variation of membrane in the long time run. How dose the operation parameters determeted? Does the UF anf MF process need washing? Therefore, I think the munuscript should be revised carefully before considering acceptance

1.     The value of Conductivity in Table 3 should be checked, in which the conductivity of retentate was much lower than that of permeate. It was unreasonable.

2.     There was NF in the Fig.3, and it was describe as a part of the whole process in line 144. However, I didn’t find any results of NF. Is it tested?

3.     It was mentioned in line 157-158 that the RO was carried out on the post-UF permeate until a four-fold reduction in the feed. Did it mean that the volume of retentate solution after the treatment of RO was a quarter of the one of feed, and the recovery ratio of RO was 75%? But I think the concentration of rententate in table 3 was not inconsistent with this recovery ratio.

Author Response

20.02.2023

Dr inż. Anna Kowalik-Klimczak

Response on review 2

about manuscript:

„Circular Economy Approach in Treatment of Galvanic Wastewater Employing Membrane Processes”, Anna Kowalik-Klimczak, Anna Gajewska-Midziałek, Zofia Buczko, Monika Łożyńska, Maciej Życki, Wioletta Barszcz, Tinatin Ciciszwili, Adrian Dąbrowski, Sonia Kasierot, Jadwiga Charasińska, Tadeusz Gorewoda

Thank you very much for the valuable suggestion and insightful comments contained in the review. Answers to the comments are presented below:

Ad. This manuscript proposed a integrate membrane process for the treatment of galvanic  wastewater. It was a attractive work for the industrial application of membrane process. However, much more details and investigation should be presented in the manuscript. For example, the fouling the flux variation of membrane in the long time run. How dose the operation parameters determited? Does the UF and MF process need washing?

Author response:

As part of the work, a demonstration of the possibility of using UF and RO processes for galvanic wastewater treatment was carried out. The processes were carried out periodically with the use of sewage with a volume of 100 dm3. In the next stage of work, tests are planned to verify the possibility of using UF and RO processes to run continuously. These tests will be carried out in real operating conditions in the company. They will make it possible to assess the possibility of implementing this technology in the industry.

In the works carried out on a semi-technical scale, no significant decrease in the permeate flux was observed in the UF and RO processes. After the processes, the membranes were rinsed with demineralized water, which enabled the return of the permeate flux to its original value. This means that no components of the filtered liquid were deposited on the membranes during the UF and RO processes.

Ad. 1. The value of Conductivity in Table 3 should be checked, in which the conductivity of retentate was much lower than that of permeate. It was unreasonable.

Author response:

Thank you for your suggestion, all results have been checked and corrected.

Ad. 2. There was NF in the Fig.3, and it was describe as a part of the whole process in line 144. However, I didn’t find any results of NF. Is it tested?

Author response:

Nanofiltration was not used for these studies. Figure 3 presented the entire diagram of the integrated membrane system enabling operation in various configurations of unit processes. However, taking into account the reviewer's suggestions, Figure 3 (now Fig. 2) has been improved.

Ad. 3. It was mentioned in line 157-158 that ‘the RO was carried out on the post-UF permeate until a four-fold reduction in the feed’. Did it mean that the volume of retentate solution after the treatment of RO was a quarter of the one of feed, and the recovery ratio of RO was 75%? But I think the concentration of retentate in table 3 was not inconsistent with this recovery ratio.

Author response:

The initial volume of the galvanic wastewater was 100 dm3. The UF was carried out until a five-fold reduction in the feed (i.e. until 80 dm3 of the permeate was removed). The RO was carried out on the post-UF permeate until a four-fold reduction in the feed (i.e. until 60 dm3 of the permeate was removed). Due to our errors in unit conversion, all obtained test results have been checked and corrected.

Reviewer 3 Report

Title: Circular Economy Approach in Treatment of Galvanic Wastewater Employing Membrane Processes

Journal: Membranes (ISSN 2077-0375)

Manuscript ID: membranes-2219063

After carefully evaluation. I am pleased to send you some comments. Please consider these suggestions as listed below to prepare the article again.  

  1. The title seems ok.
  2. The abstract seems very OK. Please add introductory lines in the beginning.
  3. Keywords are ok but maximum should be 5.
  4. Research gap should be delivered on more clear way with directed necessity for the future research work.
  5. Introduction section must be written on more quality way, i.e., more up-to-date references addressed.
  6. The novelty of the work must be clearly addressed and discussed, compare previous research with existing research findings and highlight novelty.
  7. What is the main challenge?
  8. The introduction is massive in the present form
  9. What is problem statement? State clearly.
  10. Do not use lumpy references such as 1-5. Maximum should be 2 or 3. Please revise your paper accordingly since some issue occurs on several spots in the paper.
  11. Please check the abbreviations of words throughout the article. All should be consistent.
  12. Please remove reference 1-5 and simply cite this single reference- Yaqoob, Asim Ali, et al. "Advanced Technologies for Wastewater Treatment." Green Chemistry for Sustainable Water Purification (2023): 179-202.
  13. The main objective of the work must be written on the clearer and more concise way at the end of introduction section.
  14. Reference 21 and 22 are also irrelevant at page 2 Line 74. Please cite these two references here. (a) Potential use of ultrafiltration (UF) membrane for remediation of metal contaminants (b) Application and fabrication of nanofiltration membrane for separation of metal ions from wastewater.
  15. There are several irrelevant reference please take a strong revision in this section.
  16. Please provide space between number and units. Please revise your paper accordingly since some issue occurs on several spots in the paper.
  17. What is about data repeatability?
  18. Please provide high quality image of figure 4 and 7.
  19.  Each section of results does not have scientific output why?
  20. To meet the journal standard, author should add a comparative profile.
  21. Conclusion section is missing some perspective related to the future research work, quantify main research findings, and highlight relevance of the work with respect to the field aspect.
  22. To avoid grammar and linguistic mistakes, MAJOR level English language should be thoroughly checked. Please revise your paper accordingly since several language issue occurs on several spots in the paper.
  23. Reference formatting need carefully revision. All must be consistent in one format. Please follow the journal guidelines.  

Decision = Major revision. It was tough for me to read and go through, but I was able to make a comments for improvement. Please put forth your best efforts and revised it. The idea is good so, I recommend a chance to revise it.

Author Response

20.02.2023

Dr inż. Anna Kowalik-Klimczak

Response on review 3

about manuscript:

„Circular Economy Approach in Treatment of Galvanic Wastewater Employing Membrane Processes”, Anna Kowalik-Klimczak, Anna Gajewska-Midziałek, Zofia Buczko, Monika Łożyńska, Maciej Życki, Wioletta Barszcz, Tinatin Ciciszwili, Adrian Dąbrowski, Sonia Kasierot, Jadwiga Charasińska, Tadeusz Gorewoda

Thank you very much for the valuable suggestion and insightful comments contained in the review. Answers to the comments are presented below:

Ad. 1.  The title seems ok.

Author response:

Thank you very much for your opinion.

Ad. 2. The abstract seems very OK. Please add introductory lines in the beginning.

Author response:

As recommended by the reviewer, the abstract was changed.

Ad. 3. Keywords are ok but maximum should be 5.

Author response:

Thanks for the suggestion, however, according to the instructions for authors, an article can contain 3-10 keywords.

Ad. 4. Research gap should be delivered on more clear way with directed necessity for the future research work.

And

Ad. 6. The novelty of the work must be clearly addressed and discussed, compare previous research with existing research findings and highlight novelty.

Author response:

The available literature in the field of galvanic wastewater treatment is focused on the use of membrane techniques primarily for water recovery. However, no data is available on the use of residue after filtration. The undertaken research works meet the need to use hazardous post-filtration streams as secondary raw materials in technological processes in galvanizing plants. As suggested by the reviewer, the authors supplemented the introduction section.

Ad. 5. Introduction section must be written on more quality way, i.e., more up-to-date references addressed.

Author response:

The introduction section was corrected about up-to-date references.

Ad. 7. What is the main challenge?

Author response:

The main challenge of the conducted research was the use of RO retentate as a secondary raw material in the galvanizing process. To achieve this goal, it was necessary to select the parameters of the galvanizing process and the composition of the galvanizing bath based on RO retentate.

Ad.8. The introduction is massive in the present form.

Author response:

As suggested by the reviewer, the introduction has been shortened.

Ad. 9. What is problem statement? State clearly.

Author response:

The complexity of the technology and various chemical reagents, which were used make the galvanic wastewater particularly dangerous for the environment and require neutralization. The adopted treatment methods focus primarily on water recovery. Regardless of the applicated method, the management of emerging retentates is usually not the subject of research in the available literature.

Ad. 10. Do not use lumpy references such as 1-5. Maximum should be 2 or 3. Please revise your paper accordingly since some issue occurs on several spots in the paper.

And

Ad. 12. Please remove reference 1-5 and simply cite this single reference- Yaqoob, Asim Ali, et al. "Advanced Technologies for Wastewater Treatment." Green Chemistry for Sustainable Water Purification (2023): 179-202.

Author response:

As recommended by the reviewer, all lumpy references were corrected.

Ad. 11. Please check the abbreviations of words throughout the article. All should be consistent.

Author response:

All abbreviations were checked and corrected.

Ad. 13. The main objective of the work must be written on the clearer and more concise way at the end of introduction section.

Author response:

The authors improved the purpose of the paper in the introduction section.

Ad. 14. Reference 21 and 22 are also irrelevant at page 2 Line 74. Please cite these two references here. (a) Potential use of ultrafiltration (UF) membrane for remediation of metal contaminants (b) Application and fabrication of nanofiltration membrane for separation of metal ions from wastewater.

Author response:

As recommended by the reviewer, the authors verified the references in line 74 and changed them.

Ad. 15. There are several irrelevant reference please take a strong revision in this section.

Author response:

The authors made a strong revision in the section of references.

Ad. 16. Please provide space between number and units. Please revise your paper accordingly since some issue occurs on several spots in the paper.

Author response:

As suggested by the reviewer, the authors corrected errors in the notation of values and units.

Ad. 17. What is about data repeatability?

Author response:

The study was conducted in batch UF process using 100 dm3 of wastewater. During bath membrane processes major efficiency drop was not observed. According to observation results were recognized as promising and the next stage of the study will be checking technology in real conditions directly in the company, which is interested in the implementation of those solutions. These results will be realized in real operating conditions directly on the technological line of the company, which is interested in implementing this type of solution. The tests in the factory on the technological line will allow us to optimize and increase the efficiency of technology in continuous conditions. Such action will enable the assessment of the effectiveness and efficiency of the developed technology during long-term processes.

Ad. 18. Please provide high quality image of figure 4 and 7.

Author response:

As suggested by the reviewer, the authors corrected figures 4 and 7 (now fig. 3 and 6).

Ad. 19. Each section of results does not have scientific output why?

And

Ad. 20. To meet the journal standard, author should add a comparative profile.

Author response:

As suggested by the reviewer, the authors added a comparative profile.

Ad. 21. Conclusion section is missing some perspective related to the future research work, quantify main research findings, and highlight relevance of the work with respect to the field aspect.

Author response:

As recommended by the reviewer, the authors improved the conclusion section.

Ad. 22. To avoid grammar and linguistic mistakes, MAJOR level English language should be thoroughly checked. Please revise your paper accordingly since several language issue occurs on several spots in the paper.

Author response:

As recommended by the reviewer, the text has been checked and corrected by a qualified person.

Ad. 23. Reference formatting need carefully revision. All must be consistent in one format. Please follow the journal guidelines. 

Author response:

As suggested by the reviewer, the authors have revised the section of references and corrected mistakes.

Round 2

Reviewer 1 Report

The manuscript has been revised in a logical manner, although there are still several points that may be questionable. However, the authors have adequately addressed these points, and it can be claimed that the manuscript is now ready for publication.

Reviewer 3 Report

I re-reviewed again. Well.... author addressed all concerns very carefully.  Now accepted in present form.